# On Vulnerability of Selected IoT Systems to Radio Jamming—A Proposal of Deployment Practices

**DOI:** 10.3390/s20216152

**Published:** 2020-10-29

**Authors:** Kamil Staniec, Michał Kowal

**Affiliations:** Department of Telecommunications and Teleinformatics, Wroclaw University of Science and Technology, 50-370 Wrocław, Poland; michal.kowal@pwr.edu.pl

**Keywords:** machine-type communications, ultra-narrowband, carrier to interference, packet error rate, interference, anechoic chamber

## Abstract

Weightless and SigFox are both narrowband communication systems designed for the Internet of Things, along with some other counterparts such as LoRa (Long Range) and narrowband Internet of Things (NB-IoT). As systems dedicated specifically for long-range operations, they possess considerable processing gain for energetic link budget improvement and a remarkable immunity to interference. The paper describes outcomes of a measurement campaign during which the Weightless and SigFox performance was tested against variable interference, generated in an anechoic chamber. Results allow the quantitative appraisal of the system behavior under these harsh conditions with respect to different operational modes of the two investigated IoT systems. The outcomes are then investigated with respect to an intentional radio jammer attempting to block a base station (BS) operation by directly radiating an interfering signal towards it. An Interference Margin is proposed for a quantitative expression of a system’s resilience to jamming. This margin, calculated for all available configuration settings, allows the clear assessment of which combination of a system’s operational parameters does and which does not provide immunity to this type of radio attack.

## 1. Introduction

One of the first formalized approaches to the problem of electromagnetic (EM) interference affecting Internet of Things and wireless sensor networks can be found in document [1]. It is stressed there that this issue affects the segment between the sensors/meters tier and the Access Points (AP) tier. The topic is treated in general terms in that it introduces potential victims (targets) and consequences that EM interference imposes on the Machine-To-Machine (M2M) communication. The document also presents some recommendable countermeasures.

The major aim of the narrowband Internet of Things (IoT) systems is to carry the traffic (mainly) from a large number of end-devices attached to utility meters, sensors, detectors, gauges, etc. According to the IoT traffic model presented in [2], once the IoT systems have become fully ubiquitous, the prospective number of the aforementioned sensing devices is expected to be at the level of 40 units per a household. Within the coverage of a single IoT base station (BS), it is assumed that the number of houses is 4275. Such an arrangement, though economical, creates a significant single point of failure when the BS is exposed to intensive interference or jamming, thus disabling the entire IoT network, a situation can be viewed as a form of electromagnetic cyberattack. Figure 1 illustrates this case by presenting a scenario in which readings from multiple sensors (represented by blue arrows, carrying the desirable carrier power, *C*) are sent towards the base station which at the same time is being jammed by a perpetrator emitting inband interference (I) causing “carrier to interference and noise” (CNIR), received by the BS, to drop below a threshold necessary for correct uplink (UL) reception.

The example presented in this article, albeit relating to Weightless and SigFox, is in fact generic to all kinds of narrowband IoT systems since any system performance can be defined by how its packet error rate (PER) changes with CNIR (e.g., see results for LoRa in [3]). Moreover, the limited width of the ISM (Industrial, Scientific, Medical) frequency bands in which IoT networks operate, i.e., on the order of a few hundred kHz, makes it easy to generate such a jamming signal spanning the entire band at no noticeable degradation of its power density.

## 2. The Paper Organization

The organization of the paper is as follows: Section 1 and Section 3 provide a rationale for the undertaken investigation; in particular, by explaining how an IoT BS reception can be disrupted by a single transmitter located within its coverage area, if one uses improper operational transmission settings such as the bandwidth or data rate. Section 4 gives an overview of the literature that concerns similar aspects. Section 5 introduces the technical side of two Internet of Things (IoT) systems under study, namely “SigFox” and “Weightless”, with respect to their key operational parameters. Section 6 presents outcomes of a measurement campaign carried out in a specialized laboratory. Its unique outputs quantitatively describe the SigFox and the Weightless intrinsic susceptibility to noise or interference. Section 7 contains derivation of a model for calculating the interference margin (*M_int_*) as a measure of a systems immunity to jamming, obtained for particular operational settings. Section 8 concludes the investigations in the form of practical remarks regarding the deployment of wireless sensor network BS’s in a manner assuring maximum robustness against possible radio jamming.

## 3. The Problem Statement

In this article a verification will be provided regarding the immunity of a Weightless BS and SigFox to intentional interference originating from an intentional jammer disturbing their operation. This verification will be performed in two steps: Firstly, the packet error rate (PER) response will be obtained by means of performing PER measurements in the anechoic chamber located in the Laboratory of Electromagnetic Compatibility (LEC) at Wroclaw University of Science and Technology (WUST), for all possible combinations of operational settings available in SigFox and Weightless. Secondly, knowing PER dependence on CNIR, results will be transformed into a, so called, interference margin (*M_int_*) providing a measure of a system’s resistance to jamming (Section 6.2 will provide an in-depth discussion of *M_int_*).

The said assumption of a jammer transmitting at “legal” EIRP (Equivalent Isotropic Radiated Power) levels can be considered to be a limiting factor of the finding arrived at with the use of this model, since it appears to be natural for a jamming device to radiate at the highest power available. However, as also noted in Section 8.1, the fact that an IoT network can be threatened by a relatively low-power culprit, makes the situation even more dangerous as such weak signals may be difficult to detect by means of typical search methods such as angular spectrum scanning.

## 4. Related Works

Several approaches to the problem of jamming in IoT networks have been identified in the available literature. For instance, game theory has been applied in [4] to the jamming problem in OFDM-based (Orthogonal Frequency Division Multiplexing) IoT systems by smartly distributing power among the subcarriers; this is an approach suitable for cellular IoT systems that use the orthogonal frequency division multiplexing, unlike Weightless and SigFox. In [5] a deception method is proposed in which IoT end-devices can obtain knowledge of a jammer’s activity and deceive it by sending fake messages and harvesting the jamming signal to boost its own energetic resources. However, apart from requiring considerable intelligence on the part of the device, the method is focused on the terminals, whereas a more effective way to disrupt an IoT network operation seems to be by means of attacking its gateway, as in the current paper, since individual devices are usually scattered across a large area as opposed to a BS that has a well-defined single location. In [6] the authors propose another game-based approach allowing the performance of an optimal power allocation strategy in response to a jamming action, suitable for multi-channel systems. Other sources recommend varieties of solutions to defeat radio jamming, most of which fall into one of the three categories: frequency hopping (e.g., [7,8]), power control (e.g., [9,10]) or backscatter communication, an approach originally proposed in [11] and [12] for increasing an effective transmission range and only recently recognized as an anti-jamming technique in [13] or [14].

In the authors’ opinion, all of the multiple approaches discussed above have three ideas in common: firstly, they assume that a jamming signal occupies certain distinct frequency channels (one or more). Secondly, the jamming signal is assumed to be powerful, preferably exceeding permissible levels imposed on the effective radiated power. Thirdly, they are usually abstract from any concrete system in particular, concentrating on some common technical features common to all IoT systems, with no effort placed on performing measurements on actual systems. Therefore, in the paper it was also shown that a jammer does not necessarily have to exceed permissible radiation levels to effectively jam an IoT base station, nor does it have to concentrate its power on a particular channel. Since most Low Throughput Networks (LTN) systems operate in a narrow 868 MHz frequency band, it suffices to generate the permissible EIRP = 14 dBm across the entire available bandwidth (i.e., 600 kHz for Weightless and 192 kHz for SigFox) to jam such a system, especially if it works with high data rate settings. In the paper the authors measured two real IoT systems (Weightless and SigFox) for their response to jamming in terms of PER/CNIR curves and on this basis they analyzed these systems’ performance with respect to all possible operational modes. As it turned out, the built-in transmission mechanisms and parameters, such as the modulation, coding, multiple packet repetitions, spectrum spreading or extremely low bandwidth, can be effective means of counteracting the jamming attacks. They must be, however, carefully selected to make a network operate in a trade-off between the data rate and immunity to jamming. 

## 5. Presentation of the Selected IoT Systems: The Weightless and the Sigfox

### 5.1. Systematics of IoT Systems

To date the IoT systems have achieved technological maturity and developed into two major groups and a few sub-groups within them, as shown in Figure 2. Their common feature consists of providing low-throughput, extremely long rage connectivity to the machine-type communication (MTC) traffic, i.e., one originating from multiple devices deployed across an area much vaster than available with traditional wireless technologies (such as GPRS). They differ though in the way they have originated, i.e., either as new proprietary solutions (LTN in Figure 2) or over the course of evolution of traditional cellular technologies (Cellular IoT, CIoT in Figure 2) [15,16,17]. The latter, CIoT, has spawned three alternative systems, known as Enhanced-Coverage Global System for Mobile Communications for MTC (EC-GSM-MTC), Long-Term Evolution MTC (LTE-MTC) and Narrowband IoT (NB-IoT). The systems of interest in the paper belong to two LTN sub-groups. The first sub-group, represented by Weightless, is Low-Power Wide Area Network (LPWAN), specified in [18,19,20] and characterized by bandwidths (BW) of 10–500 kHz and data rates *R_b_* of c.a. 1–100 kb/s. The other sub-group, represented by SigFox, is ultra-narrowband (UNB) [21], embracing systems whose channels bandwidths, compared with the total transmission band, satisfy a relationship of being no greater than 1:100. In SigFox with an uplink BW of 100 Hz, this ratio reaches 1:1920, since the entire band has a width of 192 kHz.

### 5.2. Weightless

The system can be viewed as one of the major players in the Low Power Wide Area Network (LPWAN) space, beside the better known LoRa [22,23]. Similar to its LoRa counterpart, Weightless, recently renamed as plain “Weightless”, also takes advantage of the spread spectrum (SS) transmission for improving system immunity. However, as opposed to LoRa (that uses multi-state chirp spreading, see also [24]), the spectrum is direct sequence spread (DSS) [25], along with other techniques, such as interleaving and data whitening. This spreading only takes place in the OQPSK (Orthogonal Quadrature Phase Shift Keying) modulation mode (MOD), whereas in the other mode, GMSK (Gaussian Minimum Shift Keying), transmission without spectrum spreading is used (i.e., *SF* = 1). It is an open communication standard dedicated to operating at long distances under harsh attenuation and interference conditions, invented and supported by Weightless SIG. It is designed for bidirectional, fully synchronized, low-power wide area public or private networks with end devices satisfying the Internet of Things (IoT) requirements such as the limited throughput and relaxed latency, fitted for operation in sub-gigahertz ISM bands. Due to legal constraints in Europe, investigations presented in the paper were performed for the ‘V band’ (863–870 MHz) ISM Band, especially its ‘M’ and ‘P’ sub-bands defined in [26], with the following limits on Equivalent Radiated Power (ERP) and duty cycle (DC):‘M’ sub-band: 868–868.60 MHz, ERP = 25 mW, DC ≤ 1%;‘P’ sub-band: 869.40–869.65 MHz, ERP = 500 mW, DC ≤ 10%.

In order to ensure a greater capacity, 12.5 kHz channels can be combined into bundles of eight channels, with the aggregate bandwidth (BW) of 100 kHz (a wideband mode, WB). Although operation in a single 12.5 kHz sub-channel (i.e., a narrowband mode, NB) implies lower data rates but increases the number of logical channels, leading to improved uplink cell capacity, a feature particularly valuable in congested networks. These two channel bandwidths, along with OQPSK and GMSK modulations, two values of coding rate (i.e., 0.5, 1) and three spreading factors (i.e., eight, four, one), define eight operational modes differing in data rates, as stated in Table 1.

### 5.3. SigFox

SigFox is a representative of the UNB systems class (see Figure 2), wherein a matter of the highest priority is the service coverage, leaving the transmission rate and on-time delivery as less significant matters. The consortium that created it [27] authorizes specific companies by providing them with BS modules and conferring on them an operator’s right for using the system. End-customers, wishing to base their own networks on SigFox, become clients to these authorized entities [28]. The remarkable energetic efficiency required from SigFox system devices finds its expression in the Random Frequency Time Division Multiple Access (RFTDMA) scheme, wherein the transmit channel is randomly picked in both time and frequency without any prior mechanisms for detecting the possible channel occupancy. Although similar to the pure ALOHA protocol, where only the time domain is unslotted, in RFTDMA also the carrier frequencies are chosen from a continuous range (192 kHz wide in SigFox) instead of being picked from a discrete set of channels as in ALOHA. A transmit spectrum of a sequence of several SigFox transmissions is shown in Figure 3. This random channel selection, beside relieving transmitting devices from any efforts associated with energy-consuming channel sensing, also allows for the use of lower quality transmitters in end-devices, even such whose local oscillator frequencies uncertainty is greater than the channel bandwidth (BW) itself, i.e., 100 Hz in uplink. Each UL packet (i.e., a single message) is retransmitted three times, as stated above, on different, randomly selected channels, prolonging the effective single packet transmission up to 6.24 s and providing 4.8 dB of extra processing gain *G_p_* to the link budget. A list of SigFox key operational features includes:the frequency range 868.034 MHz–868.226 (yielding 192 kHz of the total band);uplink (UL) bandwidth, *BW*: 100 Hz;downlink (DL) bandwidth, *BW*: 600 Hz;network topology: star;maximum EIRP: 14 dBm (25 mW);modulations: Differential Binary Phase Shift Keying (DBPSK) in UL, GFSK in DL;maximum transmission rate *R_b_* in UL: 100 b/s;maximum transmission rate *R_b_* in DL: 600 b/s;sensitivity *P_min_* equal to −144 dBm in UL and −134 dBm in DL (assuming the Signal-to-Noise Ratio, *SNR* =7 dB, *G_p_* = 4.8 dB and a noise factor (NF) equal to 5 dB and 3 dB in DL and UL, respectively).

## 6. The Measurement of the Weightless and Sigfox Susceptibility to Interference

### 6.1. The Measurement Set-Up

Electromagnetically-isolated chambers, in particular anechoic chambers (AC) and reverberation chambers (RC), create convenient environments for measuring wireless systems, by providing controlled electromagnetic background. This feature is particularly crucial when investigating systems at very high pathloss conditions, causing them to work with received signal power close to their sensitivity, at which point any unwanted signal from outside could easily disturb measurements. The usability of such artificial environments for precise investigation of wireless systems in general has been demonstrated, for example, in [29] and [30]. Demonstration of their applicability to specific wireless systems can be found in [3], [19] or [31]. Due to their superb shielding properties, chambers attenuate all external signals by c.a. 85 dB (as in our experiment) which ensures that receivers placed inside a chamber will be unaffected by any signals other than those arriving either from a transmitter or an intentional interferer (also located inside of it). To make measurements even more credible, chambers allow only antennas to be placed inside while leaving any signal-generating devices (e.g., terminals, base stations, signal generators etc.) outside, to prevent any unwanted radiation from on-board electronics from affecting results. The radiated power, in turn, is controlled with the use of variable attenuators connected to signal-generating devices.

Accordingly, in the current experiment a set-up was organized in a way presented in Figure 4 with the receive (Rx) and transmit (Tx) antennas spaced 6 m apart and the cabling connecting the actual transmitter (i.e., an end-device) and receiver (i.e., a base station), laid down under a layer of absorbers covering the AC floor. During measurements, the receiving antenna was subjected to varying level of interference generated with the use of an arbitrary signal generator AWG 70002 (by Tektronix, Beaverton, OR, USA) and radiated from a directional antenna oriented directly towards an investigated system’s Rx antenna. The radio frequency (RF) interfering signal was generated in the form of a continuous 350 kHz wide additive white Gaussian noise (AWGN) signal, with the output power *P_gen_* set to 14 dBm. At this power level the observed PER in both, Weightless and Sigfox, equaled 100%, which corresponded to totally corrupted transmission conditions. *P_gen_* was then successively decreased at 1 dB steps down to −16 dBm, in which process connectivity was being restored due to the continually growing Carrier-Noise and Interference Ratio (CNIR) resulting in PER decreasing to 0 % (i.e., a successful reception of all packets). The purpose of this campaign was to obtain curves translating PER to CNIR. These will be used in Section 4 as inputs to the jamming immunity theoretical model derived there, which takes these values as crucial parameters for estimating resilience of an IoT system to intentional jamming intended to corrupt the base station. The over-the-air measurements were set up in an anechoic chamber, as shown schematically in Figure 4. The assumed acceptable PER threshold was set to 50%, i.e., regarding a transmission as successful even if every second packed was dropped due to interference. The term “packet”, throughout the experiment, was meant as a single message containing e.g., sensor readings. Thus, in calculating PER, only those packets were considered that were eventually successfully detected and delivered to an end-user, after having been processed by error-correcting techniques available in those systems.

The Weightless devices used in experiments were Ubiik platform modules. The evaluation kit consisted of: a base station that served as a receiver interfered by the interfering signal (*I*) in the form of AWGN noise and an end-device serving as a transmitter. In each individual measurement a setting consisting of a unique combination of the aforementioned Weightless operational parameters, namely: {*BW; R_b_; R; MOD*}, was used, yielding eight combinations, the two extreme of which were discussed in Section 6.2, namely: {12.5 kHz; 0.625 kb/s; 0.5; OQPSK} and {100 kHz; 100 kb/s; 1.0; GMSK}.

The SigFox equipment, i.e., the end-device and the BS, were both Digi-Key devices, whereas the operational mode was set to the only one available in SigFox, i.e., DBPSK with automatic triple retransmissions of each packet. Since in SigFox each packet transmission actually takes place three times on different randomly selected physical channels, the need for repeating the PER test series in order to achieve the measurement stability, was thereby eliminated because repetitions were already ensured by the SigFox communication protocol. This random selection of the three physical channels for transmitting a given packet made the PER value obtained for a single test sequence equivalent to the tests performed for three different sequences of 100 packets in each.

A full compilation of operational parameters used in the measurements can be found in Table 2.

### 6.2. Discussion of Results

Measurement results of two PER profiles obtained for Weightless, representing its slowest and fastest operational modes, and a single profile for SigFox, are shown in Figure 5. As one can immediately notice, OQPSK modulation in Weightless provides a 15-decibel more robust operation than GMSK in terms of immunity to low CNIR. It allows the reception of data with PER < 50% at the level of c.a. CNIR = −15 dB but at data rates of c.a. 1 kb/s (a value still sufficient for a number of IoT applications). The configuration that proved to be the least robust to interference was the one offering *R_b_* = 100 kb/s and needed 15 dB greater CNIR to sustain PER < 50%. As for SigFox, in turn, since no spectrum spread techniques are involved, positive CNIR was required, from 4 dB up to 14 dB that turned out to be optimal for the system uninterrupted operation. The 50% PER threshold was attained at CNIR = 11 dB, the positive sign being due to the lack of signal spreading in SigFox. The resulting values of CNIR for which 50% of packets were successfully received (i.e., PER = 50%), collected for all eight Weightless modes and the one for SigFox, are shown in Figure 6. It can be seen there that all the other Weightless modes (not shown in Figure 5) perform with 50% success at CNIR between −15 dB and 0 dB, i.e., between the two extreme analyzed modes (0.625 kb/s and 100 kb/s).

It is assumed that since the jammer intends to interfere with the BS entire traffic, it has to radiate its power distributed over the whole reception frequency range (represented by the term *“band”* in Equation (3)), stretched over 600 kHz (the “M” band width) in Weightless and 192 kHz in SigFox. This means that only a fraction of the jammer total radiated power will actually affect the receiving BS, namely 0.1/192 for a SigFox BS and 12.5/600 or 100/600 for a Weightless BS. This fractional jammer occupancy is reflected in the third term of Equation (3). 

## 7. Calculations of the Interference Margin Based on CNIR (PER) Measurements

In this section a simple procedure will be demonstrated for translating results of the measured CNIR values (shown in Figure 6) into the interference margin *M_int_*, understood as a safety offset in the energetic link budget that secures reception from jamming. First, let us take Equations (1) and (2) for the basic free-space loss that relates the pathloss in *L_bf_* incurred by an electromagnetic wave as it propagates in the unobstructed 1-st Fresnel’s zone, that corresponds to a purely Line-Of-Sight (LOS) situation [32]. In this scenario, *L_bf_* is only dependent of the distance *d*, the signal center frequency *f* and the receiving antenna isotropic gain *G_i_* (the receiver being here the BS antenna, as indicated in Figure 1). The LOS assumption is highly probable because the malevolent party is interested in attaining the highest possible jamming effect on the BS under EM cyberattack.
(1)$$L_{bf}=d^{2}f^{2}{\left(\frac{4\pi }{c}\right)}^{2}\cdot {G_{i}}^{-1}\;\left[\frac{W}{W}\right]$$
(2)$$L_{bf\left[dB\right]}=32.45+20\log\left(d_{km}\right)+20\log\left(f_{MHz}\right)-G_{i\;\left[dBi\right]}$$

Each of the sensing devices connected to the BS operates within a perimeter defined by the cell range, marked as *CR* in Figure 7. For a generic IoT network it was defined in [32] as 866 m i.e., half of the Inter-Site Distance (ISD) equal to 1732 m. Provided that the jammer transmits at a center frequency 868 MHz with the maximum permissible power of 14 dBm, the interference power *I* delivered to the IoT BS will therefore equal to: 14 dBm − *L_bf_* (= 91.2 dB) + *Gi* (= 8 dBi) = −69.2 dBm, a rather high value as for the received radio signal power. This means that the lowest desired signal power *C* in the Weightless slowest transmission mode (i.e., 0.625 kb/s) can be up to 15 dB lower (see Figure 6), i.e., equal to −84.2 dBm, in order to sustain communication at PER = 50%.

As was already mentioned in Section 3, an Interference Margin (*M_int_*) will be used as an indicator of a “safety offset”, expressed in decibels, separating the useful signal from the interference power (CNIR), at which a system will still be able to operate with appropriate level of PER (here: 50%). The interference margin is considered as completely wasted as it approaches 0 dB. Once *M_int_* has dropped below 0 dB, interference will inevitably cause PER to exceed 50%, as stems from the measurements carried out in the anechoic chamber, presented in Section 6.1. The step-by-step procedure for calculating *M_int_* can be retrieved by following Equations (3) through (7), where I stands for interference power, *d_min_* stands for the minimum jammer separation from the BS such that its jamming signal pathloss equals *L_max_* at its highest.

In the experiment a verification was provided for how *M_int_* changed as the jammer moved toward the BS at 10% steps relative to the initial position, thus making the interference more critical with every step. The movement began at *CR* equal to 866 m away from BS, as the farthest point on the coverage brim, corresponding to the normalized distance *d*/*CR* of 1.0 in Figure 8 and Table 3. As could be expected, at this distance *M_int_* was the highest and positive both systems, regardless of their transmit mode (defined by a bit rate). As the jammer was approaching the BS site (decreasing *d*/*CR* down to 0.1), *M_int_* fell to c.a. 10 dB for the two slowest Weightless modes (i.e., 0.625 kb/s, 1.25 kb/s), whereas for the modes 6.25–10 kb/s it fell below 2 dB. For the two fastest Weightless modes (i.e., 50 kb/s, 100 kb/s), however, *M_int_* dropped below −10 dB. In SigFox, in turn, positive *M_int_* was maintained all the way from *d*/*CR* of 1.0 through 0.1, although falling to 1.8 dB at the normalized distance of 0.3. Based on these outcomes respective recommendations will be formulated in Section 8.1 regarding proposed transmit modes in Weightless in order to make the system the most robust against jamming.
(3)$$I=EIRP-L_{bf}+G_{i}-10\log\left(band/BW\right)$$
(4)$$P_{min}=I+CNIR$$
(5)$$L_{max}=EIRP-P_{min}$$
(6)$$d_{min}=10^{\frac{L_{max}-32.45-20logf\left[MHz\right]+G_{i}}{20}}$$
(7)$$M_{int}=20\log\left(2d_{min}/CR\right)$$

## 8. Conclusions

### 8.1. Practical Recommendations

The paper presents results of a measurement campaign on two leading IoT systems representing different families of the LTN group, i.e., Weightless and SigFox. Measurements were conducted in a laboratory of electromagnetic compatibility that allowed to create a controlled propagation environment to precisely control the amount and type of interference to which a measured system was exposed. Firstly, results of measurements performed in electromagnetically isolated conditions of an anechoic chamber were shown to demonstrate CNIR profiles vs. PER. A threshold PER of 50% was set to be indicative of the boundary reception quality, above which communication was deemed to have failed. The values for which PER = 50% was achieved, individually determined by means of measuring each of the Weightless eight operational modes and for SigFox uplink channel, were then transformed into an interference margin *M_int_*, a measure providing information on the system immunity to interference. Assuming a jammer radiating at a legal EIRP of 14 dBm, it was moved towards the BS location (at the cell center) at regular steps, each time recalculating *M_int_* to show how different operational modes respond to these changes. Some of the most important practical takeaways include:the “slowest” Weightless modes that use OQPSK modulation, particularly narrowband (0.625/1.25 kb/s) are at the same time the most robust in terms of resistance to jamming. Thus, these two data rate settings can be recommended for use in networks for their immunity to “legal-jamming”, as one particularly problematic to detect since the jammer’s signal does not exceed lawful limits, which makes it practically untraceable, as opposed to strong jammers;on the far end of immunity lie the two fastest Weightless modes, i.e., 50/100 kb/s, experiencing disruption in communication, expressed by negative *M_int_*, with the jammer placed at the at distances between 0.1 and 0.3 *d/CR* with respect to BS;in SigFox there exists a sufficient safety margin for jamming even at proximity between the jammer and the BS, which leaves some space for additional attenuation, e.g., due to vegetation or buildings;the idea of low throughput has proved correct for use in the Internet of Things systems. Not only because it provides low receiver noise (due to narrow channels, here: 12.5 kHz in Weightless, and 100 Hz in SigFox) but also due to increased immunity to EM cyberattacks. Their appearance is recognized to be a growing threat as the massive machine-type traffic becomes more and more prevalent in the years to come and as it conveys increasingly more crucial data concerning our living environment.

### 8.2. Further Research

Further investigations will concern the IoT systems vulnerability to the multipath effect by examining their PER response to mobile fading channels, such as Extended Pedestrian-A (EPA) and Extended Typical Urban (ETU), which models originally defined in [33] for LTE are nowadays recommended also for testing Cellular IoT systems’ performance.

## Figures and Tables

**Figure 1 sensors-20-06152-f001:**
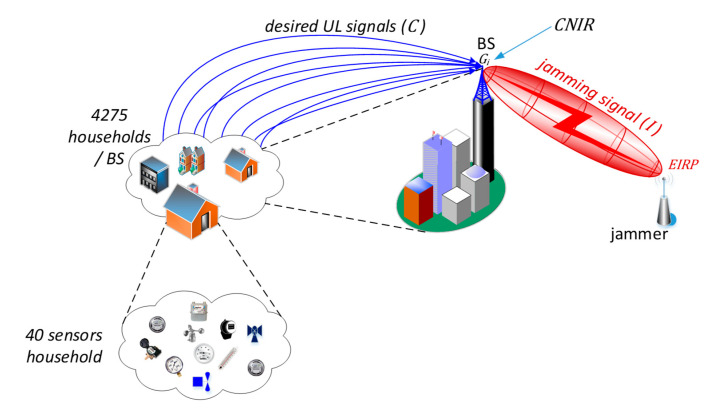
An Internet of Things (IoT) network exposed to intentional jamming.

**Figure 2 sensors-20-06152-f002:**
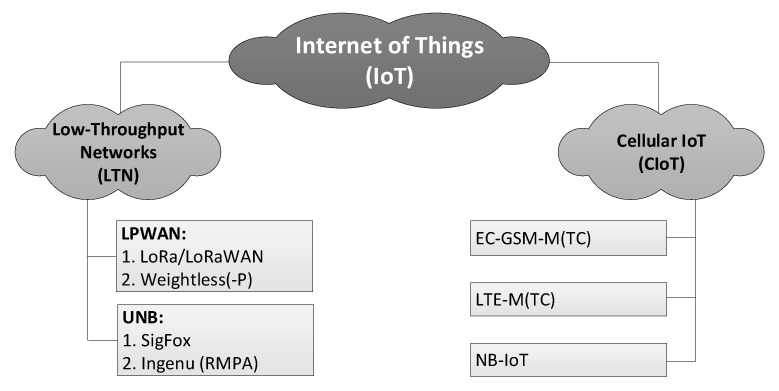
A diagram of the Internet of Things systems classification.

**Figure 3 sensors-20-06152-f003:**
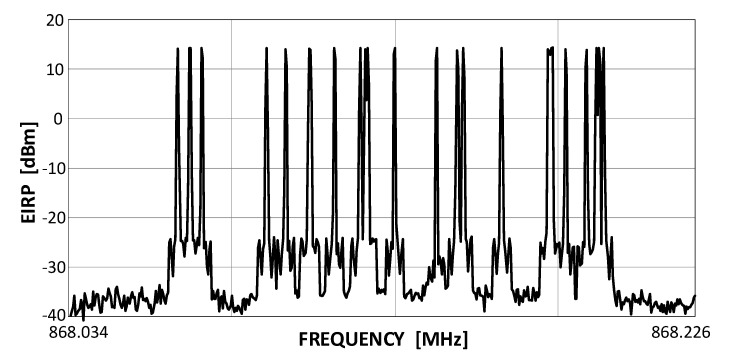
A captured output from a SigFox end-device (with EIRP = 14 dBm).

**Figure 4 sensors-20-06152-f004:**
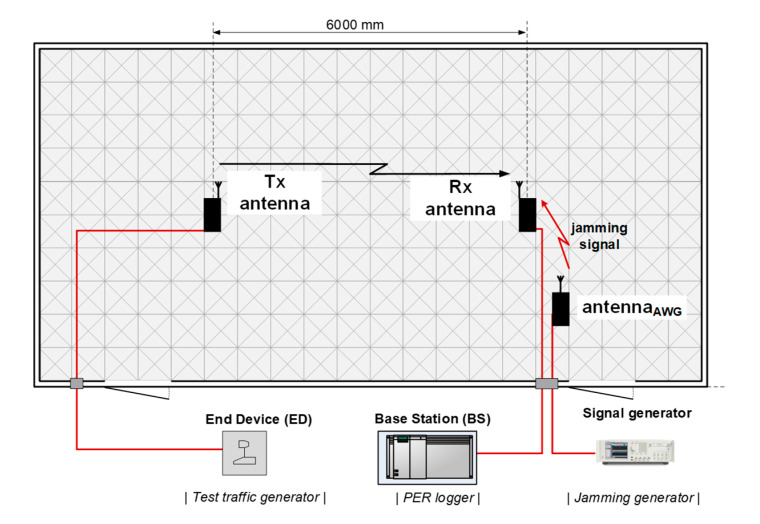
A schematic of the set-up used for measuring the IoT Low-Power Wide Area Network (LPWAN)/ultra-narrowband (UNB) systems’ performance in the Laboratory of Electromagnetic Compatibility (LEC) (Wroclaw University of Science and Technology (WUST)) anechoic chamber.

**Figure 5 sensors-20-06152-f005:**
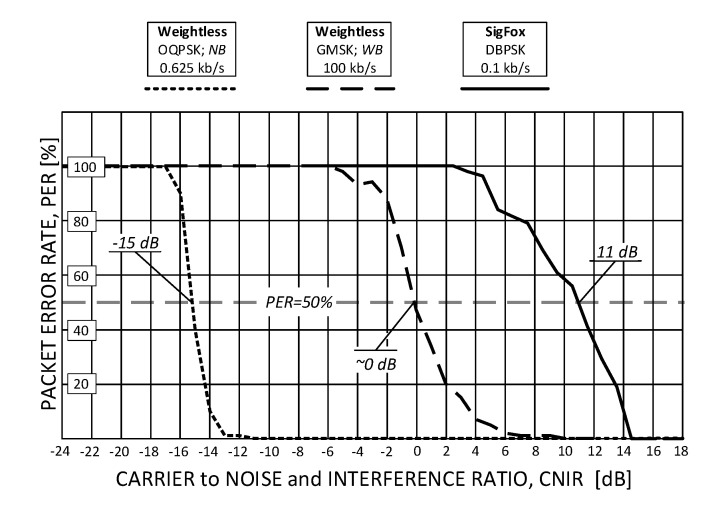
The Weightless packet error rate (PER) response to interference (jamming) expressed by carrier to interference and noise (CNIR) for two extreme operational modes.

**Figure 6 sensors-20-06152-f006:**
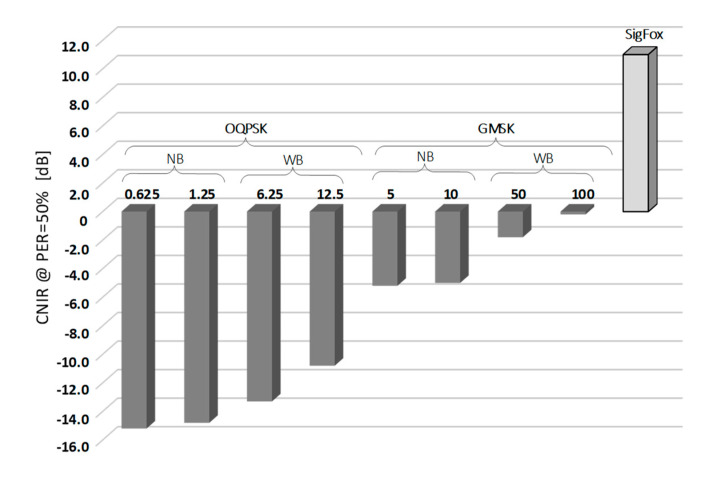
Collected measurement results of CNIR threshold measurements for achieving PER = 50% for Weightless operation modes and SigFox.

**Figure 7 sensors-20-06152-f007:**
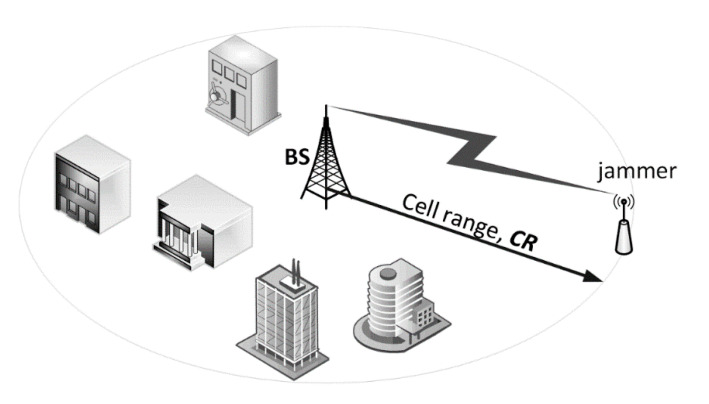
Definition of the cell range (CR) and the electromagnetic (EM) jamming situation.

**Figure 8 sensors-20-06152-f008:**
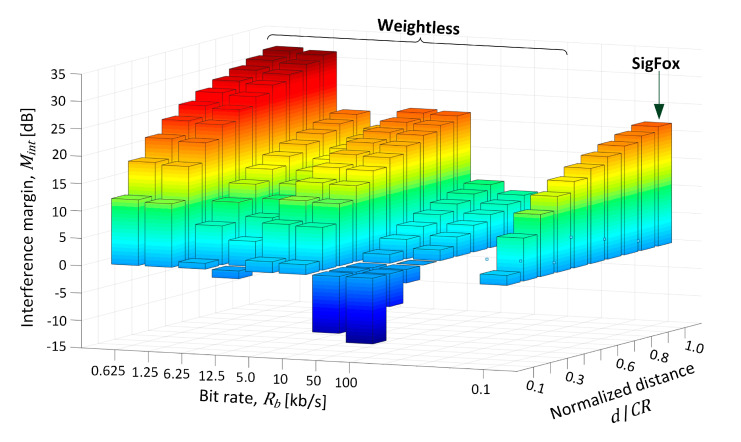
Interference margin (*M_int_*) plot for all operational modes vs. normalized distance between the jammer and the IoT base station.

**Table 1 sensors-20-06152-t001:** Relationship between the modulation, data rate and bandwidth in Weightless.

Channel Bandwidth *BW*	Modulation *MOD*	Coding Rate *R*	Spreading Factor, *SF*	Data Rate [kb/s] *R_b_*
12.5 kHz (narrowband, NB)	OQPSK	0.5	8	0.625
		0.5	4	1.25
	GMSK	0.5	1	5
		1	1	10
100 kHz (wideband, WB)	OQPSK	0.5	8	6.25
		0.5	4	12.5
	GMSK	0.5	1	50
		1	1	100

**Table 2 sensors-20-06152-t002:** A table of the set-up of the measurement.

*A Set-up Element*	*Parameter*	*Value/Type*
The wideband interference source	An arbitrary signal generator Output power, *P_gen_* Interfering signal BW	Tektronix AWG 70002 14, 13, 12… −16 dBm 350 kHz
The measurement infrastructure	Localization	An anechoic chamber
Weightless	EIRP Channel bandwidth, *BW* Center frequency	14 dBm 12.5 kHz, 100 kHz 863.1 MHz
	Data rate, *R_b_* Spreading factor, *SF* Modulation, *MOD* Coding rate, *R*	0.625 kb/s, 100 kb/s 8 and 1 OQPSK, GMSK 0.5 and 1
SigFox	EIRP Channel bandwidth, *BW* Frequency band Data rate, *R_b_* Modulation, *MOD* Coding rate, *R*	14 dBm 100 Hz 868.034–868.226 MHz 100 b/s DBPSK 1

**Table 3 sensors-20-06152-t003:** Interference margin as a function of the jammer normalized distance with respect to the Cell Range (*CR*).

$\frac{d}{CR}\;$	*Weightless*								*SigFox*
	*OQPSK*				*GMSK*				*DBPSK*
	*NB*		*WB*		*NB*		*WB*		
	*Data Rate Mode, R_b_ [kb/s]*								
	*0.625*	*1.25*	*6.25*	*12.5*	*5*	*10*	*50*	*100*	*0.1*
1.0	32.0	31.6	21.1	18.6	22.0	21.8	9.6	8.0	21.8
0.9	31.1	30.7	20.2	17.7	21.1	20.9	8.7	7.1	20.9
0.8	30.1	29.7	19.1	16.6	20.1	19.9	7.6	6.0	19.9
0.7	28.9	28.5	18.0	15.5	18.9	18.7	6.5	4.9	18.7
0.6	27.6	27.2	16.6	14.1	17.6	17.4	5.1	3.5	17.4
0.5	26.0	25.6	15.1	12.6	16.0	15.8	3.6	2.0	15.8
0.4	24.1	23.7	13.1	10.6	14.1	13.9	1.6	0.0	13.9
0.3	21.6	21.2	10.6	8.1	11.6	11.4	−0.9	−2.5	11.4
0.2	18.0	17.6	7.1	4.6	8.0	7.8	−4.4	−6.0	7.9
0.1	12.0	11.6	1.1	−1.4	2.0	1.8	−10.4	−12.0	1.8

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
