# Peer review of "On Vulnerability of Selected IoT Systems to Radio Jamming—A Proposal of Deployment Practices"

_sensors, 2020, doi:10.3390/s20216152_

Round 1

Reviewer 1 Report

Dear authors,

The paper is a good piece of work and its conclusions would be interesting for other researches. In any case, I would advice about some issues in order to improve the quality of the manuscript.

Question 1:

The introduction is clear and all the objectives well stated. At the same time, it would be beneficial for the reader if authors include an extended previous literature review to have a complete picture of the state-of-art.

Question 2:

The paper can be improved in terms of analytical model. Explain a little bit more in details what are the advantages/disadvantages of your proposal.

Question 3:

The paper could be improved in terms of application examples to other situations. Please provide some real applications in which your model would improve the vulnerability of IoT systems.

Question 4:

Please highlight the limitations of your proposal.

Question 5:

Future improvements/works could have been more discussed. Please increase this point.

Reviewer 2 Report

In this paper, the authors conduct an experiment in a controlled environment and evaluate the immunity of Weightless BS and SigFox to intentional interference.

The manuscript must be proofread carefully. The text is confusing and 

English use is not good.

As the main metric authors evaluate is the PER, I strongly suggest the authors better discuss it. How do they evaluate PER. In which network layer. Is there any packet correction technique being used?

Another major problem is that authors do not discuss results. They do not give any insight into their results. Practically any figure is explained. In some cases, they just point the figures/tables. It is mandatory that authors review and discuss the results, explaining them.

In the following, In point problems in the order of the text.

As I previously commented, the text is confusing and lacks focus. Abstract and Introduction, as important sections, must be reviewed. The subsection in the introduction does not make sense.

In the abstract

“operational parameters does” – verb

In the Introduction 

“the households density, in turn, is assumed to be 4275”

- What is the unit?

Please explain Figure 1. What is happening in this figure?

“concerns the PER response to CNIR”– define what is 

Line 61 – 

“disturbing its operation.” – their operation

line 69 

“Section 1 introduces the technical side of two Internet of Things (IoT) systems under study, namely ‘SigFox’ and ‘Weightless’ with respect to their key operational parameters”

- I did not notice this in Section 1

Line 83 - to two LTN sub-groups

Define LTN and all other acronyms in text

Line 83 – you must explain figure 2.

What is this? What are the layers?

Line 97 – “only takes place only in” – review the phrase

Line 98 – review English – “no spreading” applies to what?

Line 144 – “the former is based on proprietary solutions whereas the latter is a result of the GSM” – not clear which one you are referring to.

Line 134 – “These features,” which features? The random Chanel choosing?

Line 158 – punctuation

Line 183

You are considering this as independent event. (The three different random channel choices), which is not necessarily true.

By making three transmissions, at the same time, through three different random channels, you gain in the path- diversity. Then, this particular path-diversity increases the probability of transmission successes.

Moreover, as shown in line 185, a 100 packets sample is too low. What is the effect to increase this sample number?

How did you sample it? How did you measure? In which layer?

Line 193 - notice from where?

First, introduce the figure 5. Then, explain it.

Introduce and explain figure 6. One cannot understand it as presented.

Line 216 – translate in what?

Figure 8 and Table 2 must be discussed.

Line 251 - a measurement campaign – let clear that the measurement has been conducted in a lab, in a controlled environment.

Reviewer 3 Report

The paper “On vulnerability of selected IoT systems to the radio jamming – a proposal of deployment practices” presents an interference evaluation applied on SigFox and Weightless.

Even if the paper is interesting, in my opinion the paper has many gaps and needs to be restructured. The introduction of the paper is very long and confuses the reader. If this was the first approach presented in the scientific literature the contribution would be clear.

Also, the paper is very short (10 pages) is looks more like a review or a short communication paper than a full research article.

The presented results are promising but they need to be fully supported by real experimental data. I think the experimental work must be rework. Some details about the interference source must be added to the work.

I believe that one gap in the experimental work is the fact that these technologies are designed for far field conditions that cannot be recreated in a 6 m semi-aneconical chamber,

There are a lot of data that is not clearly presented and confusing.

Although the addressed research area is of interest, the paper is not clear on the innovation it brings. This might be due to a poor organization and description of the developed work.

 The authors must restate the main contribution of the paper.

Round 2

Reviewer 2 Report

The authors addressed all minor Issues I pointed out in the previous review round.
All metrics are now properly presented and discussed.
I also noticed extensive work to enhance the references.

Reviewer 3 Report

The quality of the paper has been incresead.